# On the Biosynthesis of Bioactive Tryptamines in Black Cohosh (*Actaea racemosa* L.)

**DOI:** 10.3390/plants14020292

**Published:** 2025-01-20

**Authors:** Martin J. Spiering, James F. Parsons, Edward Eisenstein

**Affiliations:** 1Institute for Bioscience and Biotechnology Research, University of Maryland, Rockville, MD 20850, USA; 2Fischell Department of Bioengineering, University of Maryland, College Park, MD 20742, USA

**Keywords:** black cohosh, *Actaea racemosa*, *Cimicifuga racemosa*, Rannunculaceae, secondary metabolism, serotonin, tryptophan decarboxylase, tryptamines

## Abstract

Botanical dietary supplements are widely used, but issues of authenticity, consistency, safety, and efficacy that complicate their poorly understood mechanism of action have prompted questions and concerns in the popular and scientific literature. Black cohosh (*Actaea racemosa* L., syn. *Cimicifuga racemosa*, Nutt., Ranunculaceae) is a multicomponent botanical therapeutic used as a popular remedy for menopause and dysmenorrhea and explored as a treatment in breast and prostate cancer. However, its use and safety are controversial. *A. racemosa* tissues contain the bioactive serotonin analog *N*-methylserotonin, which is thought to contribute to the serotonergic activities of black cohosh–containing preparations. *A. racemosa* has several *TDC*-like genes hypothesized to encode tryptophan decarboxylases (TDCs) converting L-tryptophan to tryptamine, a direct serotonin precursor in plants. Expression of black cohosh *TDC1*, *TDC2*, and *TDC3* in *Saccharomyces cerevisiae* resulted in the production of tryptamine. TDC1 and TDC3 had approximately fourfold higher activity than TDC2, which was attributable to a variable Cys/Ser active site residue identified by site-directed mutagenesis. Co-expression in yeast of the high-activity black cohosh TDCs with the next enzyme in serotonin biosynthesis, tryptamine 5-hydroxylase (T5H), from rice (*Oryza sativa*) resulted in the production of serotonin, whereas co-expression with low-activity TDCs did not, suggesting that TDC activity is a rate-limiting step in serotonin biosynthesis. Two *T5H-like* sequences were identified in *A. racemose,* but their co-expression with the high-activity TDCs in yeast did not result in serotonin production. *TDC* expression was detected in several black cohosh tissues, and phytochemical analysis using LC-MS revealed several new tryptamines, including tryptamine and serotonin, along with *N*-methylserotonin and, interestingly, *N*-*N*-dimethyl-5-hydroxytryptamine (bufotenine), which may contribute to hepatotoxicity. Incubation of *A. racemosa* leaves with tryptamine and *N*-methyltryptamine resulted in increased concentrations of serotonin and *N*-methylserotonin, respectively, suggesting that methylation of tryptamine precedes hydroxylation in the biosynthesis of *N*-methylserotonin. This work indicates a significantly greater variety of serotonin derivatives in *A. racemosa* than previously reported. Moreover, the activities of the TDCs underscore their key role in the production of serotonergic compounds in *A. racemosa*. Finally, it is proposed that tryptamine is first methylated and then hydroxylated to form the black cohosh signature compound *N*-methylserotonin.

## 1. Introduction

Black cohosh (*Actaea racemosa*, L.; syn. *Cimicifuga racemosa*, Nutt.) is a perennial plant of the buttercup family (Ranunculaceae) and one of the top ten selling herbs in the U.S. [1,2,3,4]. Its popularity is strongly driven by its purported positive effects on menopausal symptoms, such as hot flashes, and its apparent lack of estrogenic activity [5,6,7,8,9,10,11]. Although the exact mode of action of black cohosh-containing extracts and tablets is largely unknown and even controversial [12,13,14,15,16,17], a variety of secondary metabolites have been identified in *A. racemosa*, many of which show biological activities in vitro and in experimental animal models [1,18,19,20,21,22,23,24]. Prominent *A. racemosa* metabolites include the glycosylated cycloartenol-derived triterpenoids [25,26,27,28] and esterified phenolic dimers [23,29,30,31]. These high-abundance compounds have good utility as chemotypic markers for authentication and standardization of black cohosh preparations [1,18,22,26]. On the other hand, evidence for the involvement of, for example, triterpenoids in ameliorating menopausal symptoms is only tentative [32]. Interestingly, *A. racemosa* tissues and extracts also contain tryptamine-derived compounds, such as serotonin analogs and β-carbolines [33,34]. Of particular interest is the serotonin analog *N*-methylserotonin because it has highly potent serotonergic activity against the human 5-HT_7_ receptor [34,35], making *N*-methylserotonin a possible lead compound for explaining the mode of action of black cohosh-containing preparations in alleviating menopausal symptoms.

In efforts to elucidate the genetic basis and biochemical sequence of the production of key black cohosh metabolites, expressed sequence tag (EST) screening and homology-based cloning approaches yielded several candidate genes involved in secondary metabolism in *A. racemosa* [36]. Among these candidates are *TDC1* and *TDC2*, encoding proteins with significant similarity to tryptophan decarboxylases (TDCs), pyridoxal-phosphate (PLP)-dependent enzymes that represent the entry point into serotonin biosynthesis in plants.

TDC decarboxylates L-tryptophan to tryptamine in plants [37]. TDC is most closely related to decarboxylase enzymes that act on other aromatic L-amino acids (EC 4.1.1.28), such as L-tyrosine decarboxylase (TYDC) and L-DOPA decarboxylase (DDC). Unlike DDCs, however, which display activities toward a broad range of amino acids, TDC and TYDC have exclusive specificity for their respective substrates, L-tryptophan and L-tyrosine [37]. Tryptamine, the product of the TDC-catalyzed reaction, is a precursor compound for several plant secondary metabolites, including serotonin, β-carboline alkaloids, and indole monoterpenoids [37].

Here, it is shown that TDC1, TDC2, and TDC3 from *A. racemosa* are L-tryptophan decarboxylases that vary significantly in activity, demonstrating that black cohosh has the genes and enzymes for the production of the serotonin precursor tryptamine. The identification and in planta distribution of additional tryptamines in black cohosh tissues are also reported, and based on the results of feeding experiments, a distinct sequence of biochemical steps in the production of the different serotonin analogs in black cohosh is proposed. These studies shed light on the considerable diversity of bioactive tryptamines in black cohosh, and variable activities of the TDC enzymes may provide clues for the manipulation of tryptamine biosynthesis to improve the phytochemical quality of *A. racemosa* materials.

## 2. Results

### 2.1. TDC1 and TDC2 Are PLP-Dependent Tryptophan Decarboxylases That Differ ~Four-Fold in Activity

The first step in serotonin biosynthesis in plants is the decarboxylation of L-tryptophan to tryptamine, catalyzed by a pyridoxal phosphate (PLP)-dependent L-tryptophan decarboxylase (TDC) [38,39,40]. In previous efforts to unravel the genetic basis of serotonin production in black cohosh, two *TDC*-like genes, *TDC1* and *TDC2*, in *A. racemosa* (GenBank Accessions GU254282 and GU254283) were identified [36]. In order to test if TDC1 and TDC2 possess TDC activity, they were expressed in microbial hosts. Also included in this analysis was *TDC3* (GenBank accession HF558291), a gene closely related to *TDC1* and *TDC2*, whose transcript was isolated from black cohosh inflorescence tissues. The *A. racemosa TDCs*, when cloned into several variants of the bacterial expression vector pET22b (see Materials and Methods), yielded neither detectable soluble expression of TDC protein in *E*. *coli* cell extracts nor accumulation of tryptamine, the expected product of TDC activity.

Because the black cohosh TDCs could not be functionally expressed in *E. coli*, they were expressed from the galactose (Gal)-inducible pYES2 vector system in the eukaryotic host *Saccharomyces cerevisiae*. As shown in Figure 1, the expression of *TDC1*, *TDC2*, and *TDC3* in *S. cerevisiae* resulted in the production of tryptamine, which was not detected in the vector-only control. The media from the *TDC*-expressing cultures typically reached concentrations of ≥1.0 µM tryptamine within 24 h, demonstrating that the three *A. racemosa* TDCs had L-tryptophan decarboxylase activity. Interestingly, TDC1 and TDC3 had about four-fold higher activity than TDC2 (Figure 1). This difference was not caused by differences in yeast growth: OD_600_ values of 20-h Gal-induced yeast cultures transformed with *TDC1*, *TDC2*, or *TDC3* ranged from 1.8 to 2.6 and showed no correlation with any of the activity differences among the three TDCs. Moreover, measurements with cell extracts supported the relative differences in TDC activity identified in the whole-cell assay: TDC1 activity in extracts from 16-h Gal-induced cultures was 0.24 ± 0.02 pmol tryptamine mgFW^−1^ min^−1^, significantly higher than the activity of TDC2, which was 0.05 ± 0.01 pmol tryptamine mgFW^−1^ min^−1^ (means and standard deviations of two separate experiments; *p* < 0.05, *t*-test). To test if TDC1 and TDC2 had activity towards structurally similar aromatic amino acids, 5-hydroxy-L-tryptophan or L-tyrosine was added to TDC-transformed and Gal-induced yeast cultures, as well as to protein extracts from these cultures. Although LC-MS analysis enabled the ready detection of as little as 100 nM serotonin and tyramine, neither of the two predicted decarboxylated products of 5-hydroxy-L-tryptophan and L-tyrosine was detected in any of the TDC extracts or supernatants from TDC-expressing cultures grown 16–48 h. Thus, TDC1 and TDC2 did not exhibit detectable activity against 5-hydroxy-L-tryptophan or L-tyrosine.

Both TDC1 and TDC2 contain a highly conserved PLP-binding signature motif [36]. This region includes a lysine residue at positions 316 and 314 in TDC1 and TDC2, respectively, predicted to be required for the binding of PLP through a Schiff base. To test for this requirement, K^314(316)^ was replaced with R^314(316)^ through site-directed mutagenesis. As shown in Figure 1, when *TDC2^K314R^* was expressed in *S. cerevisiae*, no tryptamine was detected in these yeast cultures; expression of *TDC1^K316R^* likewise resulted in loss of tryptamine production. These results demonstrated that K^314(316)^ is essential for activity and supported the hypothesis that TDC1 and TDC2 are PLP-dependent enzymes.

### 2.2. A Single Amino Acid Substitution Accounts for the Difference in Activity Between TDC2 and TDC1/TDC3

The fourfold difference in the activities of TDC1 and TDC3 relative to the activity of TDC2 was surprising given the high level of overall similarity (>97%) in sequence among the black cohosh TDCs ([36] and Figure 2). As can be seen in Figure 2, an alignment of the inferred TDC protein sequences indicated that out of a total of 16 polymorphic sites, only three, at positions 222, 315, and 473 of the alignment, were uniquely present in TDC2. Of particular interest was a TDC2-specific polymorphic site within the active region—at amino acid 323 (relative to the full-length sequences of TDC2, corresponding to position 325 in TDC1)—which was a Cys/Ser variable residue. A comparison of these regions with other plant TDC enzymes revealed that the most prevalent residue at the 323/325 position was Cys, with some TDC containing a Thr, but none containing a Ser residue. To examine whether the Cys/Ser^323/325^ polymorphism contributed to the variation in activity between TDC2 and the other two TDCs, site-directed mutagenesis was used to interchange the Cys/Ser residues between TDC1 and TDC2. The resulting mutants, TDC1^C325S^ and TDC2^S323C^, were expressed in yeast and their activity was measured; all differences reported in the following section were statistically significant at *p* < 0.01 (ANOVA, Tukey’s test). As can be seen in Figure 3A, tryptamine accumulation in cultures of TDC1^C325S^ was significantly lower than in the wild-type TDC1 cultures and similar to tryptamine accumulation in the TDC2 cultures. The opposite pattern was observed for TDC2^S323C^: its expression resulted in a higher accumulation of tryptamine in cultures than in the TDC2-expressing cultures and resulted in tryptamine levels similar to those in TDC1- and TDC3-expressing cultures. These results demonstrated that the Cys/Ser^323/325^ variable site in TDC1/TDC3 and TDC2 was the main reason for the difference in activity between TDC2 and the other two enzymes. Interestingly, there were additional effects of other amino acid substitutions at this site; as shown in Figure 3B, substitution with threonine increased the activities of both TDC1 and TDC2.

Although the activity differences of wild-type and mutated *A*. *racemosa* TDCs in yeast cultures and extracts appear attributable to the Cys/Ser^323/325^ polymorphism, it is possible that the activity differences were influenced by changes in the stability of the enzymes. Initial attempts to conduct a time course of TDC activity in extractive assays were inconclusive because of the low activity of TDC2, precluding reliable measurements of tryptamine at early time points. Therefore, since tryptamine readily accumulated in media of induced yeast cultures, a whole-cell assay approach was used to establish time courses of enzyme activity. Cultures of high- and low-activity *TDCs* (TDC1 and TDC2^S323C^, and TDC2 and TDC1^C325S^, respectively; because TDC1 and TDC3 had very similar levels of activity, TDC1 was chosen as the representative of the two enzymes) were induced with galactose (Gal) for 16 h, transferred into high-glucose medium to rapidly repress TDC expression [41], and media aliquots periodically removed from these cultures to measure tryptamine. The cultures displayed very similar time courses, showing a doubling of tryptamine concentration about every 1 h in all cultures over the entire time course. Taken together, these results indicated that the difference in tryptamine accumulation between the high- and low-activity TDCs reflected intrinsic differences in enzyme activity and was not the result of differences in enzyme stability.

### 2.3. Co-Expression of TDCs from A. racemosa with Tryptamine-5-Hydroxylase (OsT5H) from Rice (O. sativa) Results in Serotonin Production

Having identified the activity for the first step in serotonin biosynthesis in *A. racemosa*, decarboxylation of L-tryptophan to tryptamine, we sought to identify gene candidates in black cohosh for the putative next step, hydroxylation of tryptamine at C-5 by tryptamine-5-hydroxylase (T5H). In rice (*O. sativa*), T5H was identified as a cytochrome P450 monooxygenase enzyme [38,42]. Thus, OsT5H was sought as a control to develop an approach for identifying potential T5H genes in black cohosh. Because expression of *OsT5H* has so far been achieved only in animal or bacterial cells [38,42], we first tested whether *OsT5H* can be functionally expressed also in *S. cerevisiae*. The *OsT5H* full-length coding sequence was PCR amplified from *O. sativa* (see Materials and Methods) and cloned and expressed in yeast. *OsT5H* was co-expressed with the black cohosh high- and low-activity TDCs. As shown in Figure 4, the production of serotonin was detectable only in yeast cultures that expressed OsT5H with the high-activity enzymes TDC1 or TDC2^S323C^. In contrast, co-expression of OsT5H with the low-activity TDCs, TDC2 or TDC1^C325S^, did not result in serotonin production. When tryptamine (at 0.1 mM concentration) was added to TDC2–OsT5H co-expression cultures, production of serotonin ensued, indicating that the absence of serotonin production in the TDC1^C325S^–OsT5H cultures was not due to lack of OsT5H activity in this particular combination. These results demonstrated functional expression of *OsT5H* in yeast and that the production of tryptamine by TDC activity appears to be a rate-limiting step in the yeast system.

With a yeast system in place to analyze T5H activity, a homology-based PCR approach as previously described by Spiering et al. [36] was used to clone *T5H*-like sequences from *A. racemosa*. More than 10 candidate sequences were identified whose predicted protein sequences had significant (e-values < 10^−5^) similarity to OsT5H. Because of this large number of T5H-like sequences, we focused only on those having >40% identity to OsT5H. Full-length coding regions of the selected *T5H*-like sequences were obtained by genome walking or by sequencing of cDNA from flowering tissues, which have high levels of serotonin (see below). This approach yielded *T5H*-*like1* and *T5H*-*like2* genes (GenBank accessions HF558292 and HF558293), with inferred protein sequences that had between 42–44% identity with OsT5H and 49% identity with each other. Like *OsT5H*, both *T5H*-*like* genes contained one intron, and their amino acid sequences showed similarity to the conserved domains in cytochrome P450 monooxygenase enzymes. However, when *T5H-like1* and *T5H-like2* were expressed in yeast, either alone with added tryptamine or co-expressed with high-activity *TDCs*, no serotonin production was detected, suggesting that T5H-like1 and T5H-like2 lacked T5H activity.

The lack of T5H activity could result from the failure of the black cohosh T5H-like enzymes to interact functionally with the yeast cytochrome P450 reductase (CPR), which is required for P450 activity. This possibility was addressed by expressing the two *T5H*-like genes in the yeast strain Wat11, engineered to express the *ATR1* gene, encoding a CPR gene from *A. thaliana* [43], and also in yeast cells co-transformed with a putative *CPR* gene from black cohosh (*CPR1*; GenBank accession HF558290). However, co-expression of the *T5H*-like genes with the two *CPRs* also failed to yield detectable serotonin. Replacement of the N-terminus of T5H-like1 with the N-terminus of OsT5H predicted to include a signal anchor peptide and truncation of the 37 and 34 N-terminal amino acids in T5H-like1 and T5H-like2, respectively, also failed to yield T5H activity. As shown below, *N*-methyltryptamine appears to be hydroxylated to *N*-methylserotonin in planta. To test for the possibility that the T5H-like proteins use *N*-methyltryptamine as a substrate, 1 mM *N*-methyltryptamine was added to yeast cultures expressing the two T5H-like enzymes from *A. racemosa*. However, this also did not result in accumulation of *N*-methylserotonin in these cultures or in cultures expressing OsT5H from rice, suggesting that neither the T5H-like enzymes from black cohosh nor OsT5H accepts *N*-methyltryptamine as a substrate for hydroxylation.

### 2.4. At Least Four Different Tryptamines Are Detectable in Black Cohosh

As shown by Powell and co-workers, the tryptamine derivative *N*-methylserotonin is an active constituent in rhizome/root tissues of *A. racemosa* [34]. It was therefore of interest to determine if *N*-methylserotonin or structural relatives could be detected in the fresh tissues of black cohosh plants. Above- and below-ground materials from *A. racemosa* plants, including rhizome/root, leaf/petiole, and flower heads, were extracted, and the extracts were analyzed by LC-MS. As shown in Figure 5A and Table 1, *N*-methylserotonin could be detected in most tissues. In addition, serotonin, not previously reported from *A. racemosa*, was also identified in these tissues. In fact, serotonin was the major tryptamine derivative detected in flowering tissues (Table 1). Both *N*-methylserotonin and serotonin displayed very large differences in concentration across tissues and also among individual tissue samples. Concentrations of both *N*-methylserotonin and serotonin peaked in flowering tissues, the only tissue type in which tryptamine was also detected (Figure 5A and Table 1). *N*-methyltryptamine, a possible precursor to *N*-methylserotonin (see below), was not detected in any tissue. Interestingly, a peak with a positive-ion mass within <10 ppm of the mass for the serotonin derivative 5-hydroxy-*N*,*N*-dimethyltryptamine (ES+ = 205.1341) eluted immediately after the peak for *N*-methylserotonin in extracts from some aerial tissues. As seen in Figure 5B, this peak co-migrated with an authentic standard of 5-hydroxy-*N*,*N*-dimethyltryptamine (bufotenine) and was tentatively identified as *N*,*N*-dimethylserotonin. The peak for *N*,*N*-dimethylserotonin was only seen in a few aerial tissues and was largely undetectable in root and rhizome tissues, suggesting a low but measurable abundance in black cohosh (Table 1).

### 2.5. Feeding Experiments Suggest Biosynthetic Routes of Serotonin Metabolites in Black Cohosh

In light of the presence of several tryptamines in black cohosh tissues and that tryptamine is a likely precursor to serotonin, it was of interest to see if tryptamine exogenously supplied to black cohosh leaves would affect the *in planta* levels of serotonin and also of *N*-methylserotonin. Because we also considered the possibility that *N*-methyltryptamine might be a direct precursor to *N*-methylserotonin, *N*-methyltryptamine was also included in this precursor-feeding analysis. Tryptamine and *N*-methyltryptamine were added to water at a final concentration of 0.1 mM in a detached leaf assay. After incubation, precursor-fed leaves and unfed (water-only) control leaves were extracted and analyzed by LC-MS. Both tryptamine and *N*-methyltryptamine could be detected within 24 h in the fed leaf tissues, indicating ready uptake of both compounds into the leaves. After 48 h of incubation, black cohosh leaves with exogenously supplied tryptamine had a concentration of >1 ppm of serotonin; in contrast, serotonin was not detected in the unfed control leaves or the leaves fed with *N*-methyltryptamine (Figure 6A). Interestingly, the concentration of *N*-methylserotonin was unchanged in tryptamine-fed leaves but strongly increased in the leaves fed with *N*-methyltryptamine (Figure 6B). To check for the possibility that chemical changes during extraction and analysis had caused the above results, extracts from the control leaves were spiked with tryptamine or *N*-methyltryptamine, but the levels of serotonin or *N*-methylserotonin were unaltered in these spiked extracts. These results indicated that the increases in the concentration of serotonin or *N*-methylserotonin in tryptamine- or *N*-methyltryptamine-fed leaves were due to conversions in the intact leaf tissues. Exogenously supplied serotonin (at 0.1 mM concentration) also accumulated in leaf tissues, but did not increase *N*-methylserotonin, supporting the observation in Figure 6B that serotonin is not methylated to *N*-methylserotonin in these tissues. Taken together, these results provide evidence for the conversion of tryptamine to serotonin and *N*-methyltryptamine to *N*-methylserotonin in black cohosh. Figure 7 summarizes the pathways to the different serotonin analogs identified in *A. racemosa* as suggested by the results of the above metabolite-feeding experiments.

### 2.6. Tissue Levels of Tryptamines Are Only Weakly Correlated with TDC Gene Expression

To examine if expression of the TDC genes was correlated with production of the various newly identified tryptamines, *A. racemosa* tissues were extracted and analyzed for serotonin, *N*-methylserotonin, and tryptamine, and *TDC* gene expression was measured by real-time reverse-transcription PCR (RT-qPCR). Because the *TDC* genes in black cohosh are highly similar in sequence, it was not possible to design primer sets specific for each *TDC* homolog, so the measured *TDC* expression could not differentiate between them. Figure 8 shows levels of *TDC* expression in the various tissues and the distribution of tryptamines in the same tissues. Expression of *TDC* was highest in leaf tissues and in seedlings and ~3–10-fold lower in belowground tissues in early spring. Surprisingly, *TDC* expression was low in flowering tissues, despite the high tryptamine levels in these tissues. Accordingly, when all tissues were included in linear regression analysis, no significant correlation between tryptamine concentration and *TDC* expression was detected (R^2^ = 0.004; *p* > 0.810). On the other hand, a relatively weak but statistically significant correlation (R^2^ = 0.411; *p* < 0.02) was detected when the flowering tissues were removed from the regression. Overall, expression of the *TDC* genes was quite low in the plant: the average C_T_ in the qPCR assay for *TDC* expression for all plant tissues of 25.5 was much higher than the average C_T_ of 16.0 for the normalizing gene *EF1α*. These data suggest that expression of the *TDCs* was more than 100-fold lower than expression of *EF1α*, which is consistent with previous observations of undetectably low expression of *TDCs* in *A. racemosa* [36].

## 3. Discussion

Black cohosh (*A. racemosa*) is a very popular dietary supplement for women experiencing menopausal discomforts. Although the active principles in *A. racemosa* responsible for ameliorating menopausal symptoms are unknown, the recent discovery of the potent serotonin receptor agonist *N*-methylserotonin in black cohosh tissues suggests serotonergic activities as one possible mode of action. Accordingly, identifying the major serotonergic compounds, their in planta distribution, and the genetic basis for their production will significantly advance our understanding of how, when, and where these compounds are made in *A. racemosa*. In this study, it has been shown that TDC1, TDC2, and TDC3 from black cohosh are PLP-dependent tryptophan decarboxylases for the production of the immediate serotonin precursor tryptamine. Additionally, it has been shown that a single amino acid substitution in the TDC active site is responsible for large differences in activity among the three TDCs, which has potential implications for serotonin levels in *A. racemosa* tissues since TDC activity appears to be a rate-limiting step in serotonin production. The results of this study have also provided evidence for several additional tryptamine compounds in black cohosh tissues and suggest a novel pathway to methylated serotonins in *A. racemosa*.

Identifying the genetic and physiological factors that control the concentration of bioactive serotonins in *A. racemosa* will provide pivotal information for improving and assuring the phytochemical quality of black cohosh materials. Of particular interest was the first committed step in serotonin biosynthesis, the conversion of L-tryptophan to tryptamine, catalyzed by tryptophan decarboxylase (TDC). By expressing TDC1, TDC2, and TDC3 in cultures of the yeast *S. cerevisiae,* it has been shown that all three enzymes have TDC activity (Figure 1). TDC1, TDC2, and TDC3 had decarboxylase activity only against L-tryptophan, in keeping with observations on other plant TDCs [37], and this strongly suggests the involvement of at least some of the three TDCs in the production of serotonin and its analogs in black cohosh.

As mentioned above, serotonergic activities in black cohosh have been invoked as one mode of action for improving menopausal symptoms, and so targeted alteration of the phytochemical profiles of serotonins may be very useful to test this hypothesis and may also enable approaches for improving the phytoconstituents in *A. racemosa*. The phytochemical profiles of plants can be changed in several ways. For example, eliciting cultures with pathogen signatures [44] or genetically altering enzyme expression [45] can alter TDC expression and the concentration of tryptamine-derived metabolites in some plant species, consistent with the rate-limiting activities observed for many TDC enzymes [40,46,47] and in our co-expression experiments of the black cohosh TDCs with tryptamine-5-hydroxylase (T5H) in yeast (Figure 4). However, overexpression of *TDC* genes to increase tryptamine production in *A. racemosa* is not yet feasible because methods to genetically modify this plant are currently lacking, and *TDC* gene expression does not respond to a range of elicitors [36]. Another strategy may rely on identifying and selecting high-activity TDC enzymes. Because the three black cohosh TDCs were so similar in sequence but differed in activity, a Cys/Ser polymorphism could be pinpointed as the likely reason for this difference, which was confirmed by site-directed mutagenesis (Figure 2). To our knowledge, this is the first demonstration of using a single amino acid exchange to modify the activity of a plant TDC. It could inform approaches to alter the levels of serotonin in black cohosh, for instance, through simple genetic screening methods to identify *A. racemosa* accessions with high-activity TDCs. Modifying TDC activity has potential uses also in other plants. For example, considerable advances have been made in reengineering metabolism and enzymes in some other plant species for the production of tryptamine-derived monoterpene indole alkaloids with new bioactivities [48]. The observed effects of amino acid substitutions in the active site of TDCs described here could help inform molecular approaches aimed at increasing the production of tryptamine-derived plant compounds with improved pharmaceutical properties.

This study has significantly expanded the list of serotonergic principles in *A. racemosa*: in addition to *N*-methylserotonin previously identified, our metabolite analysis also detected serotonin and its precursor, tryptamine, in black cohosh tissues (Figure 5A and Table 1). Thus, a greater diversity of tryptophan-derived monoamines exists in *A. racemosa* than has been previously reported [33,34]. Although the concentrations of these tryptamines were very variable in the plant, all of the plant tissues surveyed here contained at least one of these serotonergic metabolites (Table 1). This analysis also indicated the presence of a compound with a mass and retention time suggesting the presence of the dimethylated serotonin analog *N*,*N*-dimethyl-5-hydroxytryptamine (bufotenine), although it was not detected in all plant tissues (Figure 5B and Table 1). The presence of the alkaloid bufotenine, a possible psychoactive compound that binds to serotonin receptors [49], is surprising, and its occurrence throughout the plant, particularly in the roots and rhizomes of black cohosh, which are typically utilized for preparations to treat menopausal symptoms, awaits more sensitive analytical methods.

Consistent with observations of TDC gene expression in other plants [37,40,50], the black cohosh *TDCs* showed differential expression across different black cohosh tissues (Figure 8), albeit at very low levels. However, this pattern of expression was only weakly correlated with serotonin levels in these tissues. The reason(s) for the absence of any strong correlation might include the translocation of the serotonins across plant tissues, which may have obscured any such correlation—as shown in the metabolite-feeding assays with detached leaves, even the relatively hydrophobic compounds tryptamine and *N*-methyltryptamine were readily translocated within *A. racemosa* tissues. And although transcription plays a significant role in regulating the levels and activity of some TDC [37], regulation of TDC activity is achieved also at the level of protein synthesis and degradation [37]. Given these previous observations and since serotonins accumulated in many black cohosh tissues that had a detectable expression of the *TDCs* (Table 1 and Figure 8), it is reasonable to infer the involvement of the *TDC* genes in the production of the serotonin analogs in black cohosh. A cell suspension culture system for black cohosh has been developed in our laboratory (Bhavneet Kaur and Eisenstein, unpublished) and may enable more facile approaches to investigate gene–metabolite relationships in this plant, which could ultimately address the question of how the different TDC homologs contribute to serotonin production in *A. racemosa*.

The genetic, biochemical, and metabolite feeding studies described here have shed light on the biosynthetic routes for the production of bioactive serotonin metabolites in black cohosh. As shown in Figure 7, following the TDC-catalyzed formation of tryptamine, hydroxylation will yield serotonin, which requires the activity of a T5H enzyme [38]. T5H is a P450 monooxygenase that has been functionally characterized only in a limited number of species [38,51], and only the T5H protein sequence from rice is currently available [38]. Using the rice OsT5H sequence information, we cloned several T5H-like sequences from *A. racemose*, and those most closely similar to OsT5H were screened for activity by functional expression in yeast. Although expression of *OsT5H* yielded the expected product, serotonin, in yeast cultures (Figure 4), expression of *T5H-like1* and *T5H-like2* from *A. racemose* did not. This result was maintained even when *T5H*-like genes were co-expressed with an *A. racemose CPR*-like gene or expressed in a yeast host optimized for the expression of P450 enzymes from plants or as protein fusions containing N-terminal sequences from OsT5H. Also tested was whether *N*-methyltryptamine was a substrate for the *T5H*-*like* genes, but coincubation with yeast cells expressing the *T5H*-like genes (or *OsT5H* from rice) also did not result in the expected product *N*-methylserotonin. Therefore, the identity of the tryptamine-hydroxylating enzymes in *A. racemose* remains unclear.

On the other hand, incubation of black cohosh leaves with tryptamine resulted in increased concentrations of serotonin, indicating that tryptamine was readily hydroxylated in these leaves (Figure 6). Interestingly, levels of *N*-methylserotonin were unchanged in the tryptamine-fed leaves but were greatly increased in leaves incubated with *N*-methyltryptamine. These observations suggested that tryptamine is hydroxylated to serotonin in black cohosh tissues. Surprisingly, they also suggested that tryptamine is first methylated to *N*-methyltryptamine, which is then hydroxylated to form *N*-methylserotonin (Figure 7). However, it remains to be established whether the conversion of the methylated or unmethylated tryptamines to their serotonin analogs is the result of the activity of substrate-specific enzymes or arises from the presence of promiscuous hydroxylating enzymes in *A. racemosa*. Further analysis of the genes and metabolites of bioactive tryptamine biosynthesis in black cohosh should clarify this and other questions.

## 4. Materials and Methods

### 4.1. Plant Materials and Growth Conditions

Sequencing of ITS rDNA and chloroplast DNA [36,52] was used to verify the species identity of all plants used in this study. Actaea racemosa plants were grown in a greenhouse as previously described [36]; a subset of plants was grown outdoors on-site to measure compound profiles and gene expression in seasonally expressed tissues.

### 4.2. Chemicals, Strains, Enzymes, General Molecular Methods, and Biological Sequences

All chemicals used were analytical grade or molecular biology grade. UPLC solvents were supplied by Fisher Scientific (Pittsburgh, PA, USA). Restriction enzymes were supplied by New England Biolabs (Ipswich, MA, USA). Authentic standards for tryptamine, N-methylserotonin (5-hydroxy-N_ω_-methyltryptamine), tyramine, L-tryptophan, and L-tyrosine were purchased from Sigma (St. Louis, MO, USA); serotonin, 5-hydroxy-L-tryptophan, and N_ω_-methyltryptamine were purchased from Fisher Scientific. *N*,*N*-dimethylserotonin (5-hydroxy-*N*,*N*-dimethyltryptamine, bufotenine) was from Cayman Chemical. PrimeStar Enzyme (Takara Bio Inc., Otsu, Shiga, Japan) was used in polymerase chain reaction (PCR) for gene cloning and site-directed mutagenesis with PCR buffer containing 1 mM MgCl_2_. Oligonucleotides used in this study are listed in Appendix A and were purchased from Integrated DNA Technologies (Coralville, IA, USA). The Escherichia coli strain TOP10F’ (Invitrogen, Carlsbad, CA, USA) was used for cloning and amplification of plasmid constructs or T/A-cloned DNA. Bacterial expression of *A. racemosa* TDC genes was performed in E. coli strains BL21(DE3) and Rosetta 2 (EMD Bioscience, San Diego, CA, USA). The S. cerevisiae strains INVSc1 (Invitrogen) and Wat11 [43], kindly provided by Dr. Sarah E. O’Connor, John Innes Centre, Norwich, UK, were used for the functional expression of plant genes. For routine subculture, yeast cells were grown at 30 °C on agar plates or in liquid media with shaking at 250 rpm.

Biological sequence data reported in this study are available from GenBank at NCBI (accession numbers HF558290–HF558293).

### 4.3. Functional Expression of Plant Genes in Microbial Hosts

TDC genes were PCR amplified with primers TDC-PET-5-NcoI and TDC-3-HindIII, digested with NcoI and HindIII, and cloned into pET22b or modified vectors as N-terminal fusions to maltose-binding protein (MBP), SUMO, NusA, or eGFP for T7-promoter-based protein expression in IPTG- or auto-induced bacterial cultures [53].

For expression in the yeast S. cerevisiae, plant genes were amplified from cloned vector constructs or cDNA with forward and reverse primers containing built-in restriction sites (for example, KpnI and XbaI at the 5′ and 3′ ends of the amplified gene, respectively; see Appendix A) for directional cloning into the pYES2 vector (Invitrogen). Primers at the 5′ end of each gene were designed to incorporate a Kozak motif of CACACAATGTCC surrounding the start codon into cloned genes for optimal protein expression in yeast. Primer design was also used to replace the terminal cysteine in TDC proteins with valine, to remove a potentially problematic thiol group at the C-terminus. To express multiple plant genes in yeast, a second pYES vector in which the URA3 marker was replaced by a bleomycin (zeocin; ble) resistance marker was constructed by overlap-extension PCR. Primers pYES2F and URA3term + Ble3′ (Appendix A) were used to PCR amplify a linear fragment of pYES2 without the URA3 marker, and primers Ble5′ F and Ble3′ R + URA3term were used to PCR amplify a ble gene cassette, including the 5′ TEF1 promoter, from vector pICZB (Invitrogen). The two DNA fragments were spliced together by overlap-extension PCR via a complete overlap in complementary sequence between the primers URA3term + Ble3′ and Ble3′ R + URA3term; the spliced full-length construct was then amplified with primers pYES2Fnest + AscI and Ble5′ nest + AscI. AscI sites in the latter two primers were used to close the PCR-amplified vector after AscI digestion and ligation with T4 DNA ligase (Invitrogen) to create vector pYES-Zeo.

Cloned constructs were transformed into yeast cells via electroporation after treatment with a lithium-acetate method [54], and transformants were selected on yeast-nitrogen base (YNB; Sigma) agar medium lacking uracil (for selection of pYES2) or on YEG with 25 µg/mL zeocin (Invitrogen) for selection of pYES-Zeo. For the dual selection of pYES2 and pYES-Zeo, cells were grown in a YNB medium having a pH of 6.8 (adjusted with Na_2_HPO_4_), without uracil, and with 25 µg/mL zeocin added.

### 4.4. Site-Directed Mutagenesis and Gene Fusions

Site-directed mutagenesis of TDC genes was performed with PCR, using forward and reverse primers [“TDC(1 or 2) PLP” primers listed in Appendix A] that incorporated the desired nucleotide changes for amplification of 5′ and 3′ gene fragments from plasmid-cloned DNA. The two fragments were joined by PCR through a complete overlap between the two internal primers and 5′ and 3′ primers for amplification with primers TDC1-pYES-Kpn-5 (for amplification of TDC1 gene constructs) or TDC2-pYES-Kpn-5 (for amplification of TDC2 constructs) and TDC-pYES-Xba-3. Gene fusions of T5H-like genes from *A*. *racemosa* and OsT5H (see below) were constructed and cloned with the same PCR method, using primers OsT5H_N-termR33_T5H 3′ and T5H-L1F_OsT5H_N-termR33 to generate internal overlaps between 5′ and 3′ regions of OsT5H and T5H, respectively, for joining and final amplification by PCR. In the resulting protein fusions, the N-terminal 43 amino acids in T5H-like1 had been replaced with the 33 amino acids at the N-terminus of OsT5H. To truncate the N-termini of T5H-like1 and T5H-like2, cloned DNAs of the two genes were used as templates in PCR with the primers T5H-L1trunc-pYES-KpnI-5 and T5H-L1-pYES-XbaI-3, or T5H-L2trunc-pYES-KpnI-5 and T5H-L2-pYES-XbaI-3; the truncated gene constructs resulted in protein products in which the N-terminal 37 amino acids in T5H-like1 and 34 amino acids in T5H-like2 were replaced with methionine. All cloned gene constructs were sequenced (Psomagen, Rockville, MD, USA) from the flanking vector regions to ensure the absence of unintended PCR-introduced mutations and confirm the introduction of mutations in targeted sites and were transformed into yeast as described above.

### 4.5. Heterologous Expression in Yeast

For protein expression in yeast, fungal cultures were grown in 3 mL YNB medium supplemented with amino acids (L-arginine, L-leucine, L-lysine, L-tryptophan, L-threonine, and L-methionine at 250 mg/L; and L-aspartic acid, L-histidine, L-isoleucine, L-phenylalanine, L-proline, L-serine, L-tyrosine, and L-valine at 125 mg/L) in 14-mL polypropylene tubes (VWR International, Radnor, PA, USA) with shaking at 250 rpm; culture media for growth of the Wat11 strain were supplemented with 20 mg/L adenine (Sigma). Cells were grown for 16–20 h in pre-cultures (YNB with 2% glucose) at 30 °C and shaking at 250 rpm; 0.3 mL of the pre-culture was used to inoculate 3 mL of galactose (Gal)-induction medium (same composition as above but with 2% galactose added instead of glucose) and cells were incubated for 16–20 h (unless indicated otherwise) at 25 °C with shaking at 250 rpm. Cell growth in cultures was measured as OD_600_. For time courses of tryptamine accumulation in pre-induced TDC-expressing cultures after glucose repression, yeast cells that had been grown in Gal-induction medium for 16 h were harvested by centrifugation and re-suspended in 3 mL high-glucose medium containing 3% glucose, 100 mM KHPO_4_, pH 7.0; ref. [55] with 1 mM L-tryptophan and grown with shaking as described above.

### 4.6. Detached Leaf Assays for Precursor Feeding

Leaves were cut at the petiole and placed in 50 mL of DI water in 50-mL Falcon tubes containing 0.1 mM of each tryptamine compound to be tested; control leaves were placed into water only. The leaves were left on the bench under fluorescent lighting and at RT. DI water was added daily to make up for any volume lost due to evaporation. Leaf material, excluding the petioles, was extracted with mortar and pestle in 50% (*v*/*v*) aqueous methanol (10 mL per g of fresh leaf tissue).

### 4.7. Enzyme and Compound Extractions from Yeast Cells or Plant Tissues

To extract TDC enzyme activity from yeast cultures, cells from a 3–5 mL overnight culture were harvested by centrifugation (5000× *g* for 10 min at 4 °C) and re-suspended in 1 mL of ice-cold extraction buffer containing 200 mM Tris-HCl, 1 mM EDTA, and 10 µL of β-mercaptoethanol [56]; the cells were broken up by three 1-min bursts of vortexing with glass beads in 1.5-mL microcentrifuge tubes, and extracts were centrifuged at 10,000× *g* for 5 min. Approx. 0.5 mL extract supernatant was transferred to a fresh tube and used in enzyme activity assays at room temperature (RT) in a total volume of 100 µL in TDC assay buffer (same composition as the extraction buffer but with a final concentration of 50 µM pyridoxal phosphate). Reactions contained 0.1 mM of the amino acid substrate (L-tryptophan, 5-hydroxy-L-tryptophan, or L-tyrosine) and were stopped by the addition of 100 µL of 1 M NaOH. Stopped reactions were filtered through 0.45-µM membranes (Nanosep MF Centrifugal Device; Fisher Scientific) for analysis by LC-MS.

To extract tryptamines from black cohosh, fresh tissues were weighed, and 10 mL of 50% (*v*/*v*) aqueous methanol per g of plant tissue was added. Tissues were homogenized with mortar and pestle, transferred to 50-mL Falcon tubes (Fisher Scientific), and centrifuged at 5000× *g* at RT. A 0.5-mL aliquot of the cleared extract was filtered as described above, transferred to LC vials, and immediately analyzed by LC-MS.

### 4.8. UPLC-MS-TOF Analysis

Culture supernatants and extracts from yeast cells or plant tissues were analyzed by liquid chromatography (LC)–mass spectrometry (MS) methods, using an Acquity UPLC–LCT Premier XE MS-time-of-flight (TOF) system (Waters; Milford, MA). One to two µL of analyte was injected into the LC system, and compounds were fractionated on a C18 Ethylene-Bridged Hybrid (BEH) column (2.1 mm × 50 mm, 1.7 µm particle size, pore size 150 Å) at 35 °C. A linear binary solvent gradient of 0 min, 99% (*v*/*v*) solvent A, 1% (*v*/*v*) solvent B; 0.5–1.5 min, 95% solvent A, 5% solvent B; 2.5 min, 80% solvent A, 20% solvent B; 3.5–4 min, 50% solvent A, 50% solvent B; 4.5 min, 10% solvent A, 90% solvent B; 4.8 min, 5% solvent A, 95% solvent B was used. Solvent A was 0.1% (*v*/*v*) aqueous formic acid (Sigma) and solvent B was acetonitrile–0.1% (*v*/*v*) formic acid, and the flow rate was 0.4 mL/min. Eluted compounds were ionized by electrospray ionization (ESI) in the mass spectrometer at 3 kV and 30 V capillary and cone voltage, respectively. Desolvation and source temperatures were 250 °C and 110 °C, respectively, with a flow rate of 300 L/h. TOF resolution was with V or W optics, and detection was in positive ion mode. A 2 pg/µL solution of leucine-enkephalin standard (Waters) was used for internal mass calibration and infused into the reference channel via computer-controlled syringe injection.

Chemical reference standards were used for the authentication of compound and mass peaks. Standards were prepared in ultra-pure water or methanol and used to make working solutions in ultra-pure water to determine lower limits of detection and quantification as well as the concentration range giving a linear response in total ion current (TIC) in the MS. Although retention times showed some variation between runs, the compounds showed the same relative differences in elution from the column in single runs; for example, a typical elution profile had retention times of 0.97 min for serotonin, 1.04 min for N-methylserotonin, 2.29 min for tryptamine, 2.45 min for N-methyltryptamine, and 1.46 min for *N*,*N*-dimethylserotonin.

The lower limit of detection was determined as the concentration resulting in a mass peak within 0.02 Da of the expected compound mass and exceeding the noise in a solvent-only (blank) injection by a factor of five. The lower limit of quantification was identified as 3 times the limit of detection. Linear regressions (R^2^ > 0.99; *p* < 0.0001) identified the concentration range giving a linear response of detection as 0.5–20 µM for tryptamine (with lower limits of detection and quantification at 0.2 µM and 0.5 µM, respectively) and 0.2–10 µM for serotonin, N-methylserotonin, N-methyltryptamine, and tyramine (lower limits of detection and quantification were 0.1 µM and 0.2 µM, respectively, for all four compounds). The mean coefficient of variation of duplicate injections was <10% for all compounds. Chemical reference standards were loaded at the beginning and end of each run, and potential carryover between injections was checked with blank injections. Ion chromatograms were viewed and analyzed in MassLynx Version 4.1; compound mass peaks were identified and extracted according to the expected *m*/*z* of the positive ion mass of the target compound. The mass accuracy was determined within the linear range of detection and was typically within 5–20 ppm for all reference standards.

### 4.9. Cloning of a Tryptamine-5-Hydroxylase Gene from Rice and of T5H-like and CPR Genes from Black Cohosh

Genomic DNA from young leaves of *Oryza sativa* cv Drubaj (kindly provided by Dr. Matt DiLeo, Keygene) was extracted as previously described for *N*,*N*-dimethylserotonin, and used in PCR with two primer sets (pYES2KpnRiceT5Hfwd and RiceT5HR2splice, and pYES2XbaRiceT5Hrev and RiceT5HF2splice; Appendix A) to amplify the two exons in the tryptamine-5-hydroxylase gene from *O*. *sativa*, OsT5H [38]; protein GenBank Acc., ABA97037]. A complete overlap between the two internal primers was used to fuse and amplify the two OsT5H exons by PCR, resulting in the full-length OsT5H coding sequence, which was digested with restriction enzymes and cloned into pYES vectors. Homology-based cloning with primers T5HdegF2 and T5HdegR2 was used to isolate T5H-like sequences from *A*. *racemosa* using a PCR protocol described by Spiering et al. [36]; the PCR products were T/A cloned with Invitrogen’s TA Cloning Kit and sequenced. The sequence Information was then used for genome walking as previously described [36] to obtain full-length T5H-like ORF sequences. For cloning and expression of T5H-like1, its two predicted exons were amplified with T5H1-pYES-KpnI-5 and T5H1R2intronsplice, and T5H1F2intronsplice and T5H1-pYES-XbaI-3 (Appendix A), and the two fragments joined by overlap PCR as described above for OsT5H. T5H-like2 was directly amplified by PCR with cDNA from flowering tissues from *A. racemosa*. A cDNA sequence whose predicted protein product was significantly similar to cytochrome P450 reductases from plants (e-value < 1^−100^, 79% identical and 88% similar to an NADPH: hemoprotein from Eschscholzia californica, GenBank acc. AAC05022) was identified in a black cohosh EST collection described earlier (Spiering et al., 2011). The gene was named CPR1, and its full-length cDNA sequence was obtained by PCR with internal CPR1 primers as well as 5′ and 3′ RACE primers and cDNA from black cohosh leaf tissues. The cDNA sequence information of CPR1 was used to design primers ArCPR1-pYES-KpnI-5 and ArCPR1-pYES-XbaI-3 for amplification from black cohosh cDNA and cloning into pYES vectors for co-expression with T5H-like genes in yeast.

### 4.10. Analysis of Gene Expression by Reverse-Transcription qPCR

Total RNA extraction from plant tissues and removal of contaminating DNA by DNase I digestion was performed as previously described [36]. Oligo-dT-primed reverse transcription of total RNA (0.2–1 µg) was conducted with ArrayScript reverse transcriptase (Applied Biosystems, Foster City, CA, USA), and the synthesized cDNAs were used in duplicate reactions in real-time PCR with Applied Biosystems’ Power Sybr Green PCR and ABI 7300 real-time PCR system. Amplified products were run on 1.5% TBE gels to check for the absence of background and the presence of bands with the expected sizes.

### 4.11. Data Analysis

Statistical analysis of data was performed in GraphPad Prism 4.0c (GraphPad Software, Inc., La Jolla, CA, USA; http://www.graphpad.com). A *p* < 0.05 for differences between means or treatments was considered statistically significant.

## 5. Conclusions

In this study, the expression of *TDC1*, *TDC2*, and *TDC3* from black cohosh in *S. cerevisiae* has shown that all three *TDC*s encode PLP-dependent tryptophan decarboxylases for the production of tryptamine, the immediate precursor to serotonin in plants. By comparing the predicted TDC protein sequences, it was identified that a polymorphic residue within the putative region containing the PLP-binding site, and site-directed mutagenesis indicated that this polymorphism was responsible for an approx. Fourfold difference in activity between TDC2 and the other two TDC enzymes. To begin to unravel the pathways to the different serotonin analogs, a high-resolution LC-MS analysis of black cohosh tissues was conducted, which revealed that black cohosh tissues contained not only the previously identified *N*-methylserotonin, but also serotonin, tryptamine, and, surprisingly, the serotonin analog *N*,*N*-dimethylserotonin (bufotenine). This finding significantly expands the list of serotonergic compounds in *A. racemosa* and suggests an even larger variety of tryptamine-derived monoamines in black cohosh than previously believed and may provide clues on the inconsistency in its application. Lastly, feeding experiments with *A. racemosa* leaf tissues indicated that tryptamine and *N*-methyltryptamine are hydroxylated to serotonin and *N*-methylserotonin, respectively, suggesting that *N*-methylation of tryptamine precedes hydroxylation in the biosynthetic pathway to *N*-methylserotonin.

## Figures and Tables

**Figure 1 plants-14-00292-f001:**
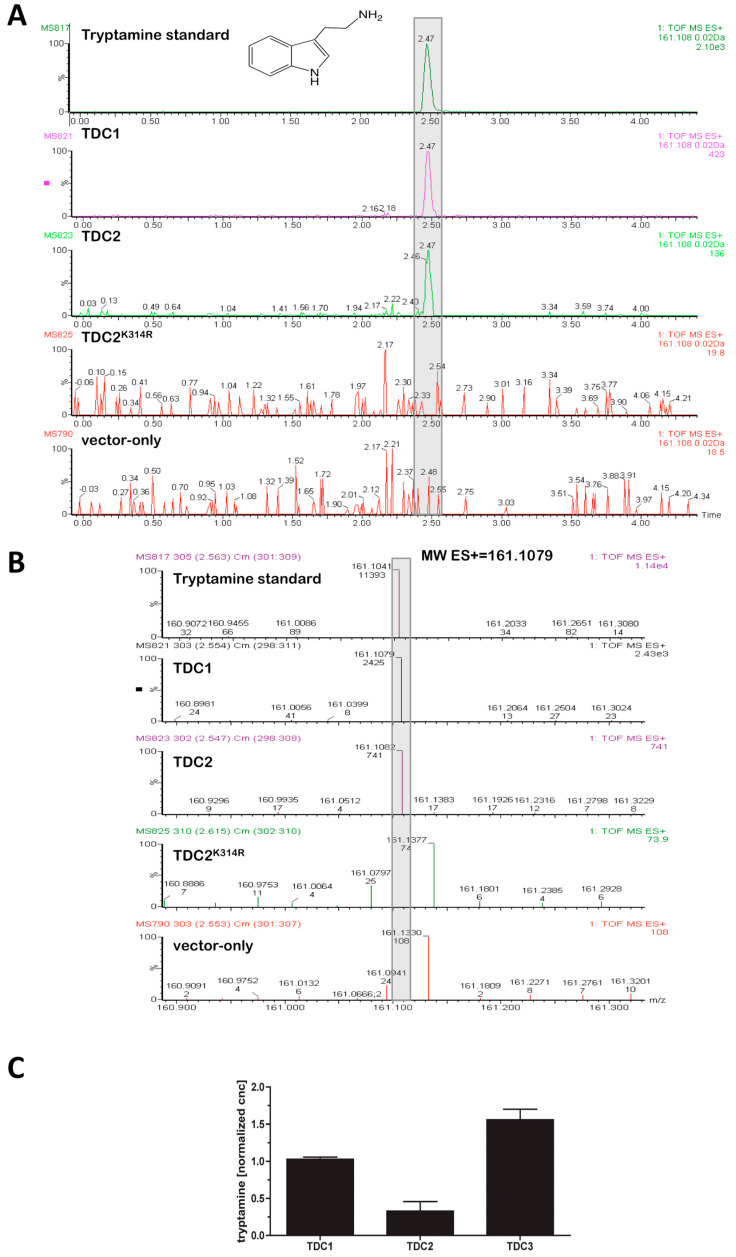
Tryptamine production in *S. cerevisiae* cultures expressing *TDC* genes from black cohosh. Panel (**A**) shows mass chromatograms of supernatants from galactose-induced 24-h cultures of yeast cells transformed with *TDC1*, *TDC2*, or *TDC^K314R^* in the pYES2 vector; cultures of yeast cells transformed with the vector only were used as a negative control. Panel (**B**) shows mass spectra extracted from the peaks boxed in grey in panel (**A**). A peak eluting at ~2.47 min, indicating a compound with a mass within 20 ppm of the expected mass of 161.1079 (in ES+ mode) for tryptamine, is present in the authentic tryptamine standard as well as in TDC1- and TDC2-expressing cultures, but not in TDC^K314R^ (in which the L-lysine residue for PLP binding was replaced with L-arginine) or the vector-only control. Panel (**C**) shows the normalized concentration of tryptamine estimated from the values for total ion current (TIC) of its mass peak in Gal-induced yeast cultures transformed with *TDC1*, *TDC2*, or *TDC3* (shown are means from two independent experiments; error bars indicate std dev).

**Figure 2 plants-14-00292-f002:**
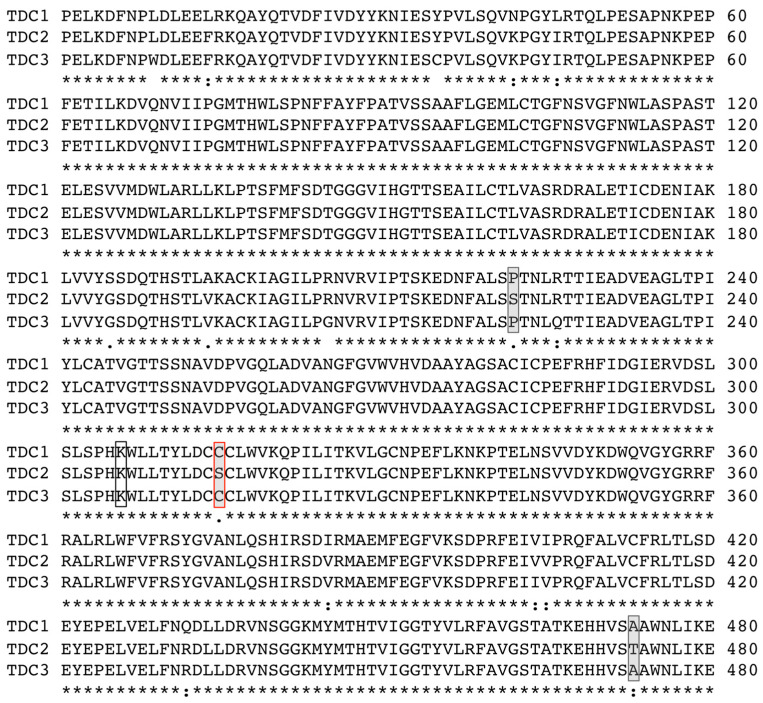
Alignment of TDC1, TDC2, and TDC3 from black cohosh. The alignment excluded 10 amino acids at the N-terminus and 9 amino acids at the C-terminus of the TDCs corresponding to a nucleic acid sequence containing the primer-annealing sites for TDC3 amplification and that, therefore, could not be unambiguously determined. A polymorphic amino acid site, representing an indel, is present within the first 10 amino acids of the N-terminus of TDC1 and TDC2 [36] and is not included in the multiple alignment; PCR results suggested that TDC3 also contains the same indel as TDC2. Amino acid residues at positions 222, 315, and 473 of the alignment are uniquely polymorphic in TDC2 and are indicated by shaded boxes; the Cys/Ser polymorphic site at position 315 (boxed with red lines) was investigated by site-directed mutagenesis (see text and Figure 3). The L-lysine residue at position 306 required for PLP binding is indicated by a clear box.

**Figure 3 plants-14-00292-f003:**
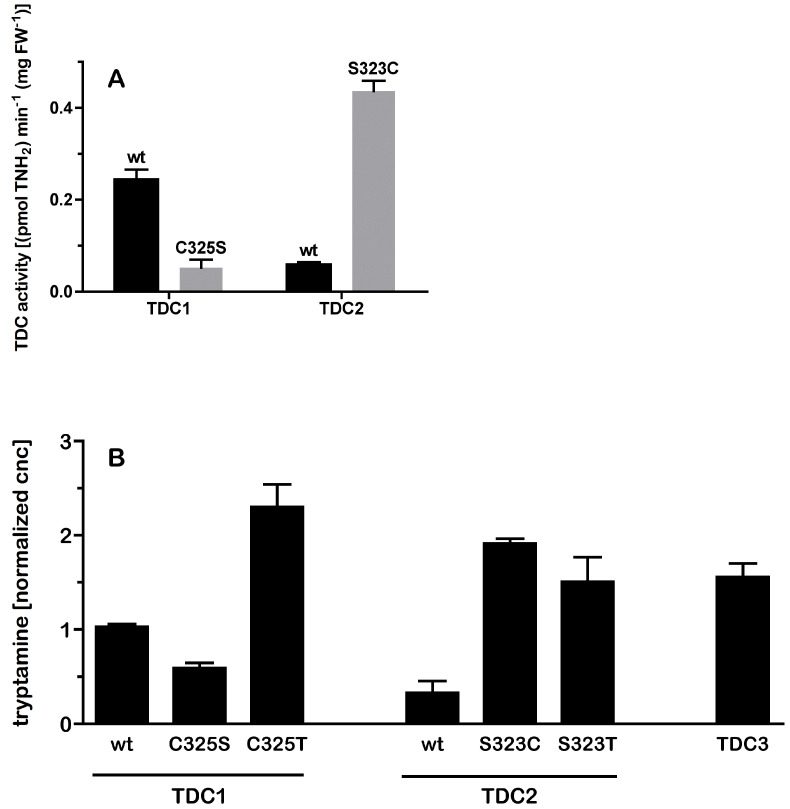
Activities of TDC1, TDC2, TDC3, and TDC variants mutated in the region of the PLP-binding active site. Panel (**A**) shows the activity of wild-type (wt) TDC1 and TDC2 (black bars) and their respective Cys/Ser PLP-site mutants (grey bars) measured as tryptamine production in yeast extracts [indicated are means and error bars show the standard deviation (std dev); n = 2]. Panel (**B**) shows the accumulation of tryptamine in yeast cultures transformed with the wild-type (wt) genes *TDC1*, *TDC2*, and *TDC3*, and mutant variants of *TDC1* and *TDC2*. Data are representative of two separate experiments and were normalized to the median of all measurements; shown are the means from two biological replicates (=independent transformants), measured in duplicate. Error bars indicate std. dev.

**Figure 4 plants-14-00292-f004:**
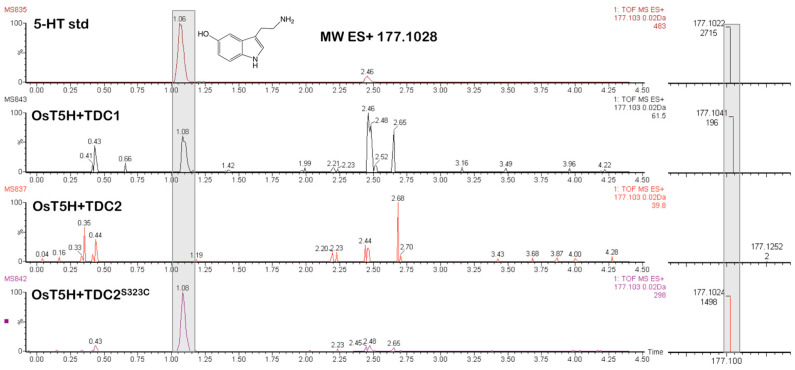
Co-expression of *TDCs* from black cohosh with *OsT5H* from *O. sativa*, encoding tryptamine-5-hydroxylase. Yeast cells co-transformed with *OsT5H* in pYES-Zeo and *TDCs* from *A. racemosa* in pYES2 were Gal-induced for 24 h, and culture supernatants were analyzed for serotonin (5-HT) by LC-MS. Mass peaks in chromatograms and in extracted mass spectra (shown to the right of each chromatogram) shaded in grey correspond to 5-HT as determined by retention times and predicted and measured mass of authentic 5-HT standard (shown at the top).

**Figure 5 plants-14-00292-f005:**
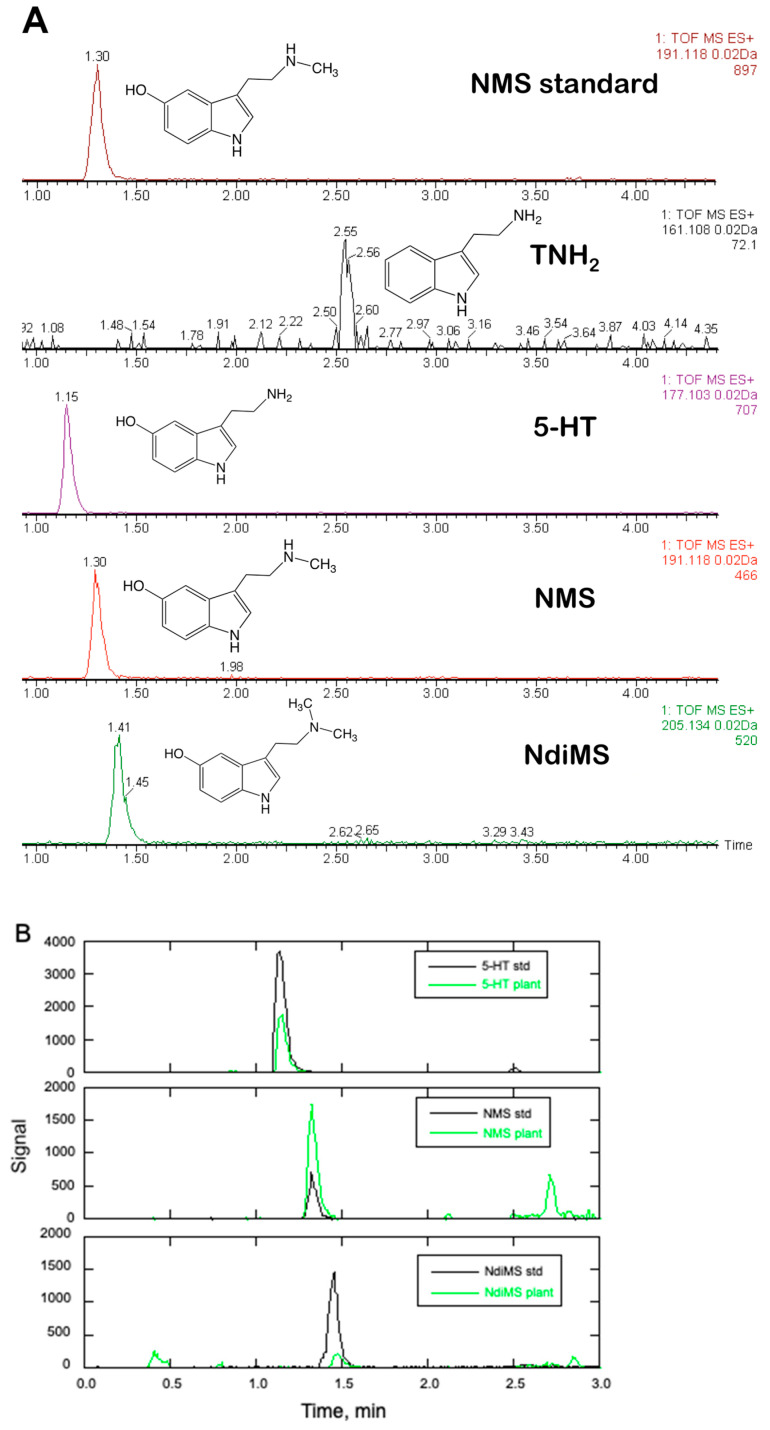
Tryptamine compounds in young flower tissues of black cohosh. Panel (**A**) shows mass chromatograms of aqueous methanol extracts of flower buds and the observed masses for their peaks of bioactive tryptamines from *A. racemosa* plants. Chromatograms are shown for an *N*-methylserotonin (NMS) standard (top), and the bioactive tryptamine (TNH_2_), serotonin (5-HT), *N*-methylserotonin (NMS), and *N*,*N*–dimethyl-5-hydroxytryptamine (*N*,*N*-dimethylserotonin or bufotenine (NdiMS). Tryptamines were identified by comparing their retention times with those of their respective authentic standards. Panel (**B**) shows overlaps of mass chromatograms of methanolic leaf extracts *A*. *racemosa* and standards for serotonin (5-HT), *N*-methylserotonin (NMS), and *N*,*N*-dimethylserotonin (NdiMS). The following masses were observed (expected masses and differences (in ppm) of expected *versus* observed mass are indicated in parentheses): NMS standard, 191.1170 (191.1184; 7.3 ppm); TNH2, 161.1095 (161.1079; 8.7 ppm); 5-HT, 177.1028 (177.1023; 2.8 ppm); NMS, 191.1185 (191.1184; <1 ppm); and NdiMS, 205.1347 (205.1341; 2.9 ppm).

**Figure 6 plants-14-00292-f006:**
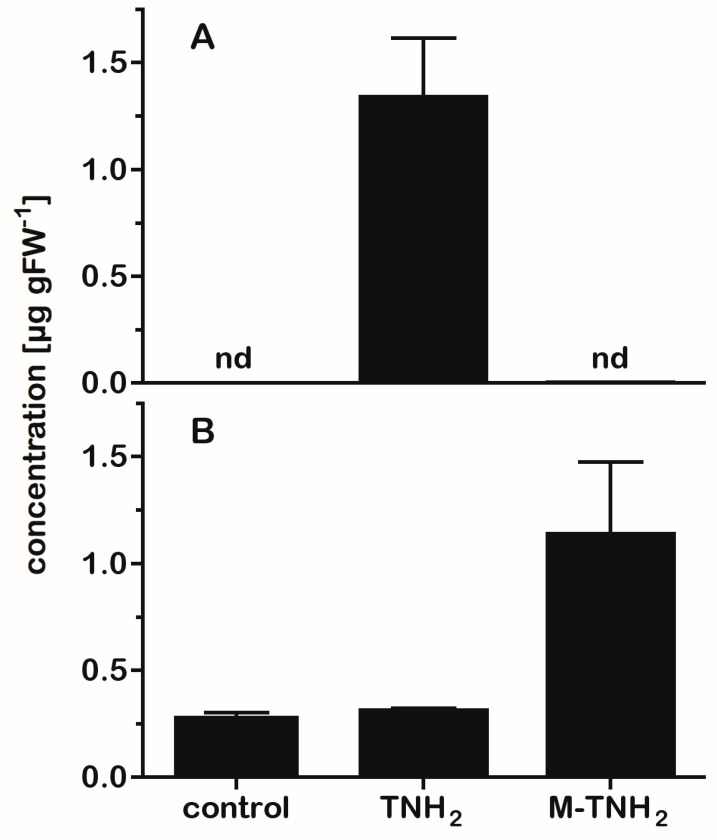
Concentrations of serotonin and *N*-methylserotonin in detached leaves incubated with tryptamine (TNH_2_) or *N*-methyltryptamine (M-TNH_2_). Leaves were incubated with 0.1 mM of TNH_2_ or M-TNH_2_ for 48 h and extracted, and concentrations of serotonin and *N*-methylserotonin were measured by LC-MS. Panel (**A**) shows the concentration (as µg per g of fresh leaf tissue) of serotonin in leaves from the control (water only) and in the leaves treated with TNH_2_ and M-TNH_2_; panel (**B**) shows the concentration of *N*-methylserotonin in leaf tissues subjected to the same treatments. Bars show means ± std dev (n = 3); nd = not detected. The concentration of *N*-methylserotonin in leaf tissues incubated with M-TNH_2_ was statistically significantly different from its concentration in the tissues incubated with TNH_2_ or in the water-only control (*p* < 0.01, one-way ANOVA with Tukey’s multiple comparison test).

**Figure 7 plants-14-00292-f007:**
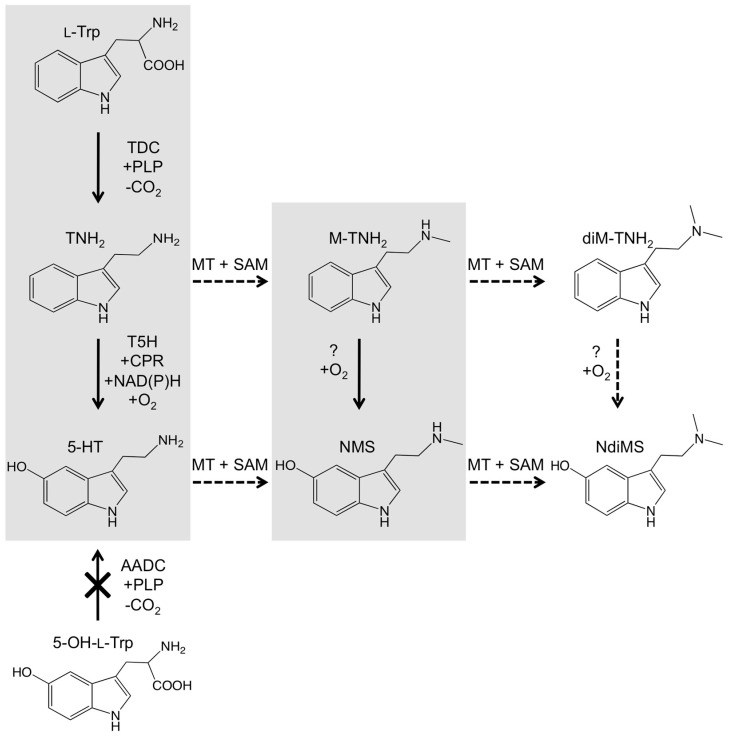
Biochemical pathways to serotonin and its methylated derivatives in plants. Solid and dashed arrows indicate confirmed and putative biochemical steps, respectively. Shading indicates biochemical routes supported by results and observations in this and previous studies. Steps with a ‘?’ symbol indicate biochemical transformations for which enzyme catalyst have not been confirmed. The shaded TDC step indicates decarboxylation catalyzed by TDC1, TDC2, and TDC3; a possible alternative route via decarboxylation of 5-hydroxy-L-tryptophan (5-OH-L-Trp) to serotonin (5-HT) has been excluded in this study for TDC1 and TDC2 and is indicated by the crossed arrow. 5-hydroxylation of tryptamine (TNH_2_) to 5-HT by a cytochrome P450 monooxygenase (T5H) has been demonstrated in rice [38]. Chemical precursor feeding results of black cohosh leaves tentatively suggest that TNH_2_ may be hydroxylated to 5-HT and *N*-methyltryptamine (M-TNH_2_) to *N*-methylserotonin (NMS), but the enzymes at these steps are unknown. Abbreviations used: AADC, aromatic amino acid decarboxylase; diM-TNH_2_, *N*,*N*-dimethyltryptamine; MT, methyltransferase; NdiMS, *N*,*N*-dimethylserotonin; SAM, *S*-adenosylmethionine; L-Trp, L-tryptophan.

**Figure 8 plants-14-00292-f008:**
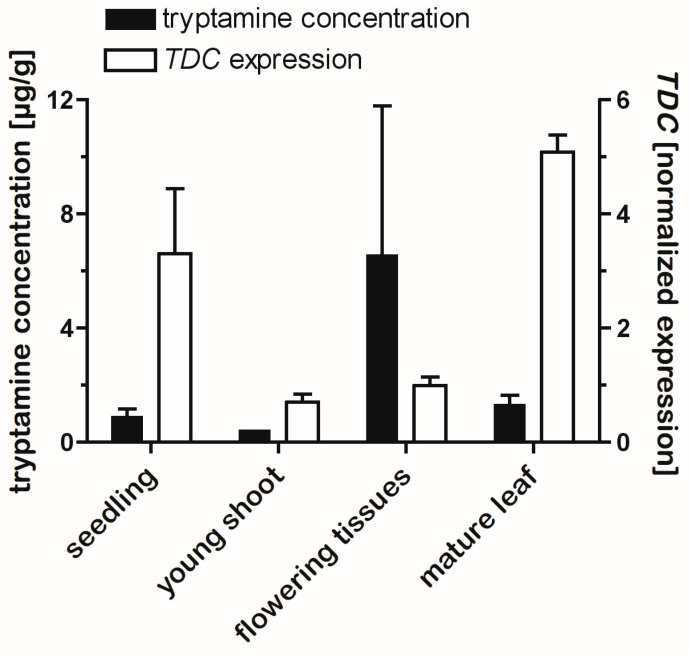
*TDC* gene expression and concentrations of tryptamines (tryptamine, serotonin, and *N*-methylserotonin) in black cohosh tissues. Solid black bars show the combined concentration of all of the three tryptamines in four different *A. racemosa* tissues (left y-axis) and open bars show expression levels of *TDC* genes measured by RT-qPCR in the same tissues (right y-axis). *TDC* expression was normalized to the expression of *EF1α* and the median of expression in all tissues. Bars show the mean ± std err (n = 3–6).

**Table 1 plants-14-00292-t001:** Concentration of bioactive tryptamines in different tissues of *A. racemosa*.

Concentration in µg gFW^−1^				
Mean (Range) ^a^				
Plant tissue ^b^	TNH_2_	5-HT	NMS	NdiMS
Rhizome	0.0 ^b^	3.7 (0.0–5.9)	5.5 (0.3–9.6)	trace (NA)
Root	0.0	0.9 (0.0–1.5)	4.1 (0.3–9.6)	0.0
Young leaf	0.0	0.8 (0.0–1.7)	0.8 (0.7–1.1)	0.0
Mature leaf	0.0	1.8 (0.0–6.9)	1.3 (0.5–4.2)	0.7 (0.0–0.7)
Flower buds	0.2 (0.0–0.4)	4.9 (0.0–8.6)	1.4 (0.0–2.7)	1.7 (0.0–3.8)
Mature flowers	0.9 (0.0–0.9)	8.6 (1.2–21.6)	4.2 (0.5–11.2)	4.4 (0.0–8.3)

^a^ Mean and range of 2–5 measurements. ^b^ Abbreviations: 5-HT, serotonin; NA, not applicable; NdiMS, N,N-dimethylserotonin; NMS, N-methylserotonin; TNH_2_, tryptamine. 0.0 = below the limit of detection (measurements given in the ranges below the detection limit were not included in the calculation of mean concentrations).

## Data Availability

The data presented in this study are available in the article.

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
