# Peer review of "On the Biosynthesis of Bioactive Tryptamines in Black Cohosh (Actaea racemosa L.)"

_plants, 2025, doi:10.3390/plants14020292_

Round 1
Reviewer 1 Report
Comments and Suggestions for Authors
Comments:
In Article “On the Biosynthesis of Bioactive Tryptamines in Black Cohosh (Actaea racemosa L.)” reported the important study of the bioactive Tryptamines in Black Cohosh. The main study reported for the TDC1, TDC2 and TDC3 from the black cohosh shows PLP-dependent tryptophan decarboxylases for production of tryptamine.
There are some following corrections recommended
1. The presentation of study is well described in this article. There are some figures in smaller size images. The figure-1 images A and B are small, there is space in the margin of page. If you make a bigger size, we can easily see the number of MW ES in spectra.
2. The Reference section does not follow the journal guidelines. The publication year in cited reference is not in bold letters, instead volume (issues) is in bold letters reported. Also, in authors’ names the semicolon is missing. Check for all the references and correct them according to the journal guidelines.
3. References number 32 and 33 are reported the same. The same references are cited two times in the reference section as well as in the descriptions section. The one of the references need to delete and correct the reference number in reference section and description sections to whole paper.
Author Response
Response to Reviewer 1 Comments
|
||
1. Summary |
|
|
We are grateful for the prompt and helpful reviews of the manuscript by Spiering, Parsons and myself submitted to the special issue of Plants on “Natural Products of Medicinal Plants.” We have revised the manuscript to address the reviewers’ comments and questions, and will summarize the responses in this letter to expedite final evaluation.
Please find the detailed responses below and the corresponding revisions/corrections highlighted/in track changes in the re-submitted files.
Thank you for your consideration,
Ed Eisenstein |
||
2. Recommended Corrections
1. 1. The presentation of study is well described in this article. There are some figures in smaller size images. The figure-1 images A and B are small, there is space in the margin of page. If you make a bigger size, we can easily see the number of MW ES in spectra.
We have increased the size of the panel to a full page (page 5, line 160), which we hope is sufficient to read the small print from the chromatograms. We explored splitting the figure into panels and showing them on two pages, but we felt that detracted from the presentation and didn’t render the information any clearer.
2. The Reference section does not follow the journal guidelines. The publication year in cited reference is not in bold letters, instead volume (issues) is in bold letters reported. Also, in authors’ names the semicolon is missing. Check for all the references and correct them according to the journal guidelines.
We have reformatted the all the cited reference in the appropriate format.
3. References number 32 and 33 are reported the same. The same references are cited two times in the reference section as well as in the descriptions section. The one of the references need to delete and correct the reference number in reference section and description sections to whole paper.
This embarrassing mistake was attributable to cutting and pasting one reference at a time from one file into the journal template file. It is now corrected, and the appropriate citation for reference 32 is now in place.
|

Reviewer 2 Report
Comments and Suggestions for Authors
The manuscript by Martin J. Spiering et al. entitled: “On the Biosynthesis of Bioactive Tryptamines in Black Cohosh (Actaea racemosa L.)” presents results on the identification and enzymatic characterization with success of three tryptophan decarboxylases (TDCs) and two putative tryptamine 5-hydroxylases in A. racemosa by their recombinant expression in yeast. This work shed some light on the rationale use of A. racemosa extracts and tablets in humans, since its purported modes of action remain largely unknown. Starting from an unexpected result, i.e. contrary to what was expected the coexpression of both A. racemosa TDC and T5H did not yield serotonin, the authors found and proposed that serotonin production in A. racemosa proceeds through N-methyltryptamine und not thru tryptamine, a very interesting and new result. The presented results are convincing. I therefore recommend this manuscript to be published in Plants since it will be of interest for the general reader by revealing a new metabolic pathway. I have only few minor questions that I would like answered by the authors in a revised version of this manuscript.
Comments
1. The data given in Fig.1 are compared by the authors, but an important information is missing. What are the amounts of A. racemosa TDC enzyme expressed in the different recombinant yeasts? The 4-fold difference could very well be due to the fact that recombinant TDC2 is produced at 4-fold lower amounts in yeast compared to TDC1 and TDC3. In that case, TDC2 would be as efficient on L-Trp than the two other A. racemosa TDCs. Could the authors comment on that? I am not speaking of enzyme stability (as presented by the authors at lines 194-196) but of levels of expression that could very well be different between the three A. racemosa enzymes.
2. Has the authors any idea of how the yeast enzymes that catabolize L-Trp by its deamination, namely Aro8p and Aro9p, interfere in their quantitative assays?
3. The authors analyze the production of the desired chemical in yeast culture supernatants. Have the authors any idea of to which extent the yeast plasma cell is permeable to serotonin? To tryptamines? Could it be that serotonin is not observed in the supernatant of coexpressed TDC2 and OsT5H would be due to serotonin being produced inside the cell but not be exported to the extracellular medium? Could the authors comment on that?
4. T5H-like proteins from A. racemosa are P450 enzymes. It may happen that a recombinant P450 is active within the cell with no product detected in the culture supernatant. The usual process for assaying activity of yeast-expressed P450 enzymes is to isolate the microsomal fractions of yeast cells and assay P450 activity on them. Did the authors try this much better approach for serotonin and methylserotonin production assays?
Minor comments
1. Abstract, line 30. Typo here, hydroxytryptamine is written: hydroxytrptamine, a “y” is missing.
2. Results, line 95. There is an error in the given GenBank accession reference for A. racemosa TDC3. The given accession (HF558292) stands for a putative T5H, whereas TDC3 accession number is HF558291.
Author Response
Response to Reviewer 2 Comments
1. Summary
We are grateful for the prompt and helpful reviews of the manuscript by Spiering, Parsons and myself submitted to the special issue of Plants on “Natural Products of Medicinal Plants.” We have revised the manuscript to address the reviewers’ comments and questions, and will summarize the responses in this letter to expedite final evaluation.
Please find the detailed responses below and the corresponding revisions/corrections highlighted/in track changes in the re-submitted files.
Thank you for your consideration,
Ed Eisenstein
2. Comments
1. 1. The data given in Fig.1 are compared by the authors, but an important information is missing. What are the amounts of A. racemosa TDC enzyme expressed in the different recombinant yeasts? The 4-fold difference could very well be due to the fact that recombinant TDC2 is produced at 4-fold lower amounts in yeast compared to TDC1 and TDC3. In that case, TDC2 would be as efficient on L-Trp than the two other A. racemosa TDCs. Could the authors comment on that? I am not speaking of enzyme stability (as presented by the authors at lines 194-196) but of levels of expression that could very well be different between the three A. racemosa enzymes.
We considered this insightful comment when conducting the original experiments. And we believe the presented data support activity differences among the TDC isozymes. Allow me to expand. Initial sequence analyses showing the differences in the active site residues (TDC1-Cys325 versus TDC2-Ser323), with the majority of plant TDCs presenting Cys at an equivalent position, we weren’t surprised by the differences in activity. And as described, an experiment to establish that the TDC1 and TDC2 enzymes displayed reasonably equivalent stability was performed. An additional, qualitative analysis of RNA via RT-PCR (using only an ethidium bromide-stained agarose gel, NOT q-RT-PCR) showed roughly equivalent levels of RNA, and hence, likely equivalent protein expression levels for enzymes that display more than 97% sequence identity. But we believe the mutagenesis experiment strongly supports the interpretation of the activity difference between the two isoforms. When Cys325 was substituted with Ser in TDC1, the measured activity was reduced to the level of the wild-type TDC2 isozyme (with Ser at the equivalent position). Alternatively, when Ser323 was substituted with Cys, it displayed an increase in activity, equivalent to that seen for wild type TDC! (with Cys at the equivalent position). The interconversion of these active site residues is good evidence that the differences in activity of the two enzymes are mainly attributable to the functional residues.
2. Has the authors any idea of how the yeast enzymes that catabolize L-Trp by its deamination, namely Aro8p and Aro9p, interfere in their quantitative assays?
Although we did not explore the role of the Aro8p and Aro9p transaminases, we did routinely look using LC-MS at the levels of supplementary additions of substrates in control cultures and found negligible decreases in their initial concentrations. Although there was doubtless some catabolism as correctly pointed out by the reviewer, it seems to have had negligible impact on the results.
3. The authors analyze the production of the desired chemical in yeast culture supernatants. Have the authors any idea of to which extent the yeast plasma cell is permeable to serotonin? To tryptamines? Could it be that serotonin is not observed in the supernatant of coexpressed TDC2 and OsT5H would be due to serotonin being produced inside the cell but not be exported to the extracellular medium? Could the authors comment on that?
A number of experimental studies in yeast have utilized substrate feeding in media to analyze heterologously expressed enzymes. Although tryptamine itself has been used in feeding experiments to produce a variety of native and specifically labeled metabolites, there is no good agreement on just how permeable tryptamines, or serotonin, are to yeast cells. However, given the fact that by adding tryptamine to yeast cells in which the expression of TDC1 and (rice) OsTH5 were induced, and that serotonin could indeed be detected in cellular assays (page 8, lines 221-224; Figure 4), suggests that at least some of the serotonin produced gets out of the cell. The observation that no serotonin could be detected in cells expressing the low-activity TDC1(Cys325Ser) or TDC2 isozymes, but was easily seen in cells with the high-activity TDC1 or TDC2(Ser323Cys) isozymes, suggest that there is some limitation (activity, metabolite concentration) that prevents detection of product under our assay conditions. Perhaps more sensitive methods would shed additional light on this question. But we believe that the presented data are consistent with the idea that TDC activity is rate-limiting under our conditions.
4. T5H-like proteins from A. racemosa are P450 enzymes. It may happen that a recombinant P450 is active within the cell with no product detected in the culture supernatant. The usual process for assaying activity of yeast-expressed P450 enzymes is to isolate the microsomal fractions of yeast cells and assay P450 activity on them. Did the authors try this much better approach for serotonin and methylserotonin production assays?
The issues of the localized expression and activity of A. racemosa P450 enzymes is indeed a key question moving forward. And, as the reviewer correctly notes, in vitro assays using microsomal fractions would indeed be useful. But this subject is the study of ongoing and future work. The fact that the heterologous expression of active TDCs and rice P450 in yeast result in serotonin being measured in culture supernatats support the elements of the pathway described. The exact components in A. racemosa that contribute to those steps awaits identification.
3. Minor Comments
1. Abstract, line 30. Typo here, hydroxytryptamine is written: hydroxytrptamine, a “y” is missing
The embarrassing spelling mistake has been corrected to hdroxytryptamine.
2. Results, line 95. There is an error in the given GenBank accession reference for A. racemosa TDC3. The given accession (HF558292) stands for a putative T5H, whereas TDC3 accession number is HF558291
The correct accession number is now used.

Round 2
Reviewer 2 Report
Comments and Suggestions for Authors
The revised version and authors' answers are correctly addresing my comments. I am now waiting the next step: microsomal activities of recombinant A. racemosa P450s!
Nice job!